# DON'T FORGET THE CONTEXT: A MULTITASK TRANSFORMER FOR INTRACORTICAL SPEECH DECODING

## ABSTRACT

We present a transformer-based sequence-to-sequence model for human speech decoding from intracortical neural recordings. Unlike prior framewise recurrent approaches trained with connectionist temporal classification, our approach jointly models neural and linguistic dynamics and generates open-vocabulary word sequences directly from the neural signal. To address the limited-data regime of human brain–computer interface datasets, we adopt a multitask framework that combines phoneme and word decoding with auxiliary supervision from Mel-frequency cepstral coefficients, and we introduce Neural Hammer & Scalpel day-specific transformation to mitigate cross-day nonstationarity. The model establishes a new benchmark in phoneme decoding on the Willett et al. dataset and improves over previous end-to-end systems in word decoding. Attention visualizations reveal interpretable temporal chunking aligned with speech segments, shedding light on emergent neural dynamics. Finally, a scaling analysis shows favorable power-law trends, suggesting that continued data growth could yield substantial gains and positioning transformers as strong candidates for future brain-to-text foundation models.

## 1 INTRODUCTION

Decoding speech from neural recordings could restore communication for individuals who have lost the ability to speak due to neurological, neuromuscular, structural, or developmental conditions, such as amyotrophic lateral sclerosis (ALS) or locked-in syndrome. Losing this ability drastically reduces quality of life and often leads to depression and isolation, making speech-oriented brain–computer interfaces (BCIs) that translate neural activity into text a clinically impactful goal. Recent advances in neuroscience and deep learning have begun to illuminate how speech is encoded in the brain (Bouchard et al., 2013; Goldstein et al., 2025; Chen et al., 2024), culminating in initial demonstrations of real-time speech neuroprostheses for communication restoration (Moses et al., 2021; Willett et al., 2023b; Card et al., 2024) (see (Chang & Anumanchipalli, 2020) for a perspective). Progress, however, remains limited by the scarcity of large, standardized human neural speech datasets. Only a handful have been released—ranging from electrocorticography (ECoG) recordings of syllables (Bouchard & Chang, 2020), through stereoelectroencephalography (sEEG) of single words in Dutch (Herff & Verwoert, 2022), to intracortical microelectrode arrays (MEAs) and full English sentences (Willett et al., 2023a; Card et al., 2025). These corpora differ in task design, procedures, and feature formats (e.g., raw voltages, spike counts, band power), complicating model development, cross-study comparison, and generalization. Compounding this, neural signals vary substantially across sessions and days, creating nonstationarities that any practical BCI must handle.

Most high-performing speech BCI decoders to date rely on recurrent neural networks (RNNs) trained with a connectionist temporal classification (CTC) objective to map neural time series to sequences of speech units. These models predict phonemes framewise and apply external n-gram language models (LMs) post hoc to generate word-level output and improve fluency (Moses et al., 2021; Willett et al., 2023b; Card et al., 2024). While effective, this hybrid approach has several limitations. First, the CTC paradigm addresses the lack of explicit temporal alignment between neural signal and speech labels by making per-frame predictions and collapsing repeats, but it does so under a conditional-independence assumption: each phoneme probability distribution is produced independently and does not condition on previously generated outputs. Second, "locality" is built in: each prediction typically attends to a narrow temporal window (80 ms in (Willett et al., 2023b)), potentially discarding long-range

cues such as slow articulatory dynamics or sentence structure. Third, the RNN encoder, even when bidirectional, has practical limits in representing very long contexts due to vanishing gradients, and it is trained separately from the downstream LM. As a result, the hybrid pipeline lacks true joint modeling of the input (neural) and output (linguistic) modalities during prediction, which may cap attainable performance and interpretability.

An alternative is sequence-to-sequence (seq2seq) modeling. Here, a decoder produces tokens autoregressively, conditioning each prediction on the entire (unmasked) input sequence and on the history of previously generated outputs. This removes the conditional-independence and locality assumptions, and allows the model to learn flexible alignments between neural evidence and linguistic units. Transformers are particularly well matched to this setting: self-attention can capture long-range, position-dependent structure; cross-attention can retrieve the most relevant neural context for each output. Indeed, the architecture of sequence-to-sequence transformers is what allowed for performance revolution in natural language processing (Vaswani et al., 2017; Devlin et al., 2019) and automatic speech recognition (Dong et al., 2018; Karita et al., 2019). In neuroAI, transformer backbones already underlie several foundation-style models (Azabou et al., 2023; Ye et al., 2025; Zhang et al., 2024; Chau et al., 2025). However, seq2seq remains underexplored for speech BCIs—particularly for open-vocabulary decoding under limited data—as reflected by the absence of seq2seq systems among top entries in the Brain-to-Text '24 challenge (Willett et al., 2024). Prior transformer decoders have largely targeted closed vocabularies ($\leq 50$ words) (Komeiji et al., 2024; Makin et al., 2020), leaving it unclear whether the advantages of high-dimensional contextual representations carry over when data are scarce and labels are unconstrained text.

In this work we develop a transformer-based seq2seq model that maps intracortical signals directly to both phoneme and word sequences. To address data scarcity and to shape useful intermediate structure, we adopt a multitask setup: (i) a transformer decoder maps neural inputs to phonemes, (ii) an auxiliary MFCC head encourages early layers to encode coarse acoustic cues, and (iii) a pretrained BART decoder introduces linguistic priors for word-level decoding. Beyond architecture, we explicitly tackle day-to-day nonstationarity with our Neural Hammer Scalpel (NHS) day-specific transform that combines a global linear alignment with a lightweight FiLM-style per-feature refinement. On the (Willett et al., 2023a) dataset, this approach improves both PER and WER relative to comparable baselines and—crucially—demonstrates that language conditioning can benefit phoneme recognition in neural BCIs. Compared to the first end-to-end neural-to-word decoder (Feng et al., 2024), our model attains better WER and provides new analyses of how language priors interact with lower-level speech units. Finally, we probe into the model's internal representations by visualizing the transformer's attention weights, shedding light on the emergent decoding mechanics.

## 2 METHODS

### 2.1 DATASET

We evaluate our approach on the intracortical speech BCI dataset from Willett et al. (2023a). This dataset consists of high-resolution neural recordings from an ALS patient with anarthria, collected via four 64-channel Utah microelectrode arrays—yielding 128 recording channels in area 6v (ventral premotor cortex) and 128 in area 44 (part of Broca's area). Only the signal from area 6v was used in the modeling, due to low correlation of the area 44 signal to speech prediction observed in previous work by Willett et al. and in the interest of reducing the computational load. During experiments the participant attempted to speak a total of 12,100 sentences spanning a large vocabulary (drawn from the Switchboard corpus), across multiple sessions over 24 days. Each sentence trial included an instructed delay period (planning) followed by a go cue for speaking. Only the post-go data were used in this study.

The dataset provides two neural features per channel in 20 ms bins: multi-unit threshold crossings (**spike counts**) and high-frequency power (**spike-band power**, >250 Hz). To reduce session effects, we z-score each feature per recording block. We use the train and test splits only, omitting the competition holdout. Targets are provided at two levels: (1) phonemes ( 39 ARPAbet via g2pE Park & Kim (2019), inter-word break <SIL>, sequence tokens <SOS>, <EOS>) and (2) words (tokenized with the BART tokenizer Wolf et al. (2020)). The dataset also includes 14-dimensional Mel-frequency cepstral coefficients (MFCCs) calculated from recorded audio (MATLAB 2022b). Although the microphone audio is unintelligible, it is time-synchronized and preserves envelope/onset cues; we

108 use MFCCs as weak acoustic supervision to regularize early encoder layers, not as precise phonetic
109 timing.

## 2.2 MODEL ARCHITECTURE

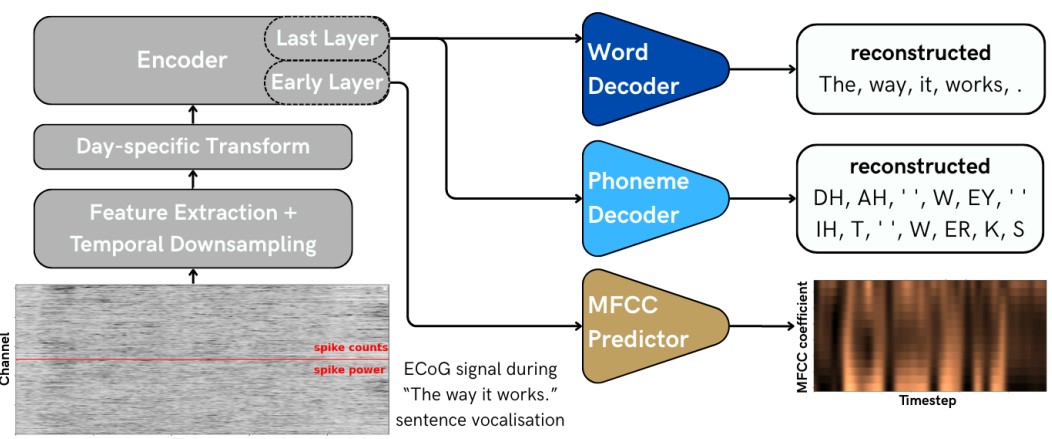

Figure 1: **Model overview.** Spike counts and spike-band power are processed in two 1-D conv branches (stride 4), concatenated, refined with wider temporal kernels and reweighted using frame-wise $1{\times}1$ content-gate. The *NHS day transform* then applies day-wise alignment by blending a *global affine* (*hammer*, cross-feature mixing) with a *FiLM* per-feature calibration (*scalpel*), combined by a learned day gate. The aligned sequence is encoded by a 6-layer Transformer and decoded by three heads: autoregressive phonemes, BART words (first 3 layers frozen), and an auxiliary MFCC predictor from encoder layer 2.

Our multitask Transformer model for neural speech decoding (Fig. 1) processes intracortical inputs through three stages—feature extraction, day-specific transformation, and a Transformer encoder—followed by three heads for phonemes, words, and MFCCs. We found training stability and performance to be highly sensitive to hyperparameters; the configuration we report aims to balance compute and accuracy and is listed in Appx. F.

**Feature extraction (content-gated, frame-wise).** Let the input be $X_0 \in \mathbb{R}^{T \times C}$ with $C{=}256$ features per 20 ms bin (128 channels $\times$ 2 features: spike counts and spike-band power). We process the two feature groups in parallel using 1-D convolutions with kernel size 3, stride 4, BatchNorm, and ELU:

$$X_1^{(1)} = \text{ELU}\big(\text{BN}\big(\text{Conv1D}_{k=3,s=4}(X_0^{\text{counts}})\big)\big), \quad X_1^{(2)} = \text{ELU}\big(\text{BN}\big(\text{Conv1D}_{k=3,s=4}(X_0^{\text{power}})\big)\big),$$

where padding is chosen so that downsampling is dominated by the stride ($s{=}4$). Each branch projects to $D/2$ channels; we use $D{=}512$. Concatenation across channels yields $X_1 \in \mathbb{R}^{\lfloor T/4 \rfloor \times D}$. A shared temporal stack integrates local context with wider receptive fields:

$$\tilde{X} = \text{ELU}\big(\text{BN}\big(\text{Conv1D}_{k=5}(X_1^{\top})\big)\big), \quad X_2 = \text{ELU}\Big(\text{BN}\Big(\text{Conv1D}_{k=7}(\tilde{X})\Big)\Big)^{\top},$$

so $X_2 \in \mathbb{R}^{\lfloor T/4 \rfloor \times D}$. Finally, a lightweight *channel-wise content gate* reweights features per frame using a $1{\times}1$ convolution followed by a sigmoid:

$$A = \sigma\big(\text{Conv1D}_{k=1}(X_2^{\top})\big)^{\top}, \qquad X_{\text{feat}} = X_2 \odot A,$$

producing a compact sequence at an 80 ms cadence (no pooling; all downsampling arises from the stride-4 convolutions). The gate is *input-driven and fast* (varies every bin) and only rescales channels (no re-mixing).

**Day-specific transformation NHS (hammer + scalpel):** To reduce cross-day drift, we apply a day-conditioned transform that mixes a global affine (*hammer*) with a per-feature FiLM modulation

(*scalpel*), combined by a learned gate and followed by a smooth non-linearity. Let $X \in \mathbb{R}^{L \times D}$ be the feature-extractor output, $d$ the day index, and $e_d \in \mathbb{R}^{128}$ a learned day embedding.

*Hammer (global affine):*

$$X_\mathrm{h} = XW_d + \mathbf{1}b_d^\top, \quad W_d \in \mathbb{R}^{D \times D}, \ b_d \in \mathbb{R}^D.$$

Each day has unique weight and bias; we initialize $W_d$ as identity plus small Gaussian noise (0.01), so the model starts near the raw space.

*Scalpel (FiLM).*

$$(\mathrm{scale}_d, \ \mathrm{bias}_d) = \mathrm{MLP}(e_d), \quad X_\mathrm{s} = X \odot (\mathrm{scale}_d \cdot 0.5 + 0.5) + \mathrm{bias}_d \cdot 0.1.$$

An MLP maps $e_d$ to unconstrained scale and bias that are applied as modulation in which the multiplicative branch is gently re-centered toward unity and the additive branch is applied with a small gain (0.1) for stability; there is no hard bounding on $\mathrm{scale}_d$.

*Gate and smooth mix:*

$$g_d = \sigma(w_g^\top e_d) \in (0, 1), \qquad \hat{X} = \mathrm{softsign}(g_d X_\mathrm{h} + (1 - g_d) X_\mathrm{s}).$$

NHS is *day-driven and slow* (constant within a trial). Via the hammer it can *re-mix* features across channels to correct global drift, while the scalpel provides mild per-channel adjustments. In ablations we also evaluate a Linear DT that keeps only the hammer (affine $W_d, b_d$; no FiLM, no gate, no nonlinearity), matching the per-day linear transform of Willett et al. (2023b).

**Encoder and Decoding Heads:** $\hat{X}$ is encoded by a 6-layer Transformer encoder ($d_\mathrm{model} = 512$, 8 heads, FFN=2048, GELU). The *phoneme head* is an autoregressive Transformer decoder trained with teacher forcing on 39 ARPAbet phonemes (+ `<SIL>`, `<SOS>`, `<EOS>`). The *word head* is the decoder-only portion of `bart-base` (Lewis et al., 2020); we freeze the first 3 decoder layers and fine-tune the remaining 3 for neural conditioning. The *MFCC head* is a linear projection from encoder layer 2 to 14 coefficients per timestep; it is optionally active in Stage 1 to shape early acoustic structure.

**Baseline architecture:** Our RNN+CTC baseline uses the same feature extractor, NHS transform, and MFCC head for parity. A Gated Recurrent Unit (GRU) RNN replaces the Transformer encoder; a linear layer predicts framewise phoneme probabilities; the BART word head decodes word-level output.

**Candidate generation and scoring:** To improve word-level decoding beyond greedy generation, we used HuggingFace's `generate()` interface to produce hypothesis set (N=148) per sentence (see Appx. B). Candidates were then rescored using three complementary signals: **PER-score:** Phoneme Error Rate between the model's own phoneme prediction and the phoneme sequence obtained by applying a G2P converter to the word-level sentence candidate. **Phoneme-score:** Mean log-likelihood of the candidate's phoneme sequence under the model's autoregressive phoneme decoder in teacher forcing, conditioned on the trial's neural encoder outputs. **LLM-score:** Sentence log-likelihood from external LM (`Mistral-7B-v0.3`), providing measure of linguistic plausibility. A linear blend of these scores, with coefficients tuned on held-out validation data, selected the final prediction for each trial.

## 2.3 TRAINING AND EVALUATION

For phoneme-only and phoneme+MFCC models, the encoder, phoneme decoder and optionally MFCC head are trained jointly for up to 300 epochs. For models including the word decoder, **Stage 1** trains the encoder and phoneme/MFCC heads for 200 epochs; **Stage 2** removes the MFCC head (if applicable), introduces the BART decoder, and continues joint training for 200 more epochs with equal phoneme and word loss weights. We use the AdamW optimizer with ReduceLROnPlateau observing the validation PER. Regularization includes dropout and simple masking augmentations (time masking up to 20% of positions, channel masking up to 15 channels).

**Losses and metrics:** Let $\mathcal{L}_\mathrm{ph}$ be cross-entropy over phoneme tokens, $\mathcal{L}_\mathrm{w}$ cross-entropy over word-piece tokens, and $\mathcal{L}_\mathrm{mfcc} = \|\hat{M} - M\|_1$ (scaled by $10^{-3}$). Stage 1 minimizes $\mathcal{L}_\mathrm{ph} + 10^{-3} \mathcal{L}_\mathrm{mfcc}$; Stage 2 minimizes $\mathcal{L}_\mathrm{ph} + \mathcal{L}_\mathrm{w}$. We report **Phoneme Error Rate (PER)** and **Word Error Rate (WER)** as normalized Levenshtein distances. PER excludes `<SOS>`/`<EOS>`; WER is computed after lowercasing and punctuation removal.

## 3 RESULTS

Table 1: Performance comparison between baseline and proposed model. Our results are reported as mean ± SD across 5 global seeds. All models besides the cited baselines are trained with the introduced Neural Hammer&Scalpel (NHS) Day Transform (DT) unless stated otherwise.

| Model | PER (%) | WER (%) |
| --- | --- | --- |
| RNN-CTC (Willett et al. 2023) 2-stage, separate LM baseline | 19.7 | **17.4** |
| BGRU-Phone (Feng et al. 2024) e2e baseline | – | 26.3 |
| RNN-CTC (Our) | $17.4 \pm 0.8$ | – |
| + MFCC | $17.0 \pm 0.3$ | – |
| + BART | $21.0 \pm 0.7$ | $30.9 \pm 0.8$ |
| + MFCC + BART | $17.0 \pm 0.5$ | $29.0 \pm 0.4$ |
| Transformer Seq2Seq | $14.8 \pm 0.3$ | – |
| + MFCC | $14.6 \pm 0.1$ | – |
| + BART | $14.5 \pm 0.3$ | $26.0 \pm 0.4$ |
| + MFCC + BART | **$14.3 \pm 0.3$** | $25.6 \pm 0.2$ |
| + MFCC + BART + generation and rescoring | $14.3 \pm 0.3$ | **$19.4 \pm 0.3$** |
| Transformer Seq2Seq + BART + Linear DT | $17.2 \pm 0.5$ | $28.6 \pm 0.4$ |
| Transformer Seq2Seq + BART + No DT | $18.1 \pm 0.6$ | $30.4 \pm 0.5$ |

Table 1 summarizes performance across baselines and ablations. At the phoneme level, the seq2seq Transformer consistently outperforms the framewise RNN–CTC family, improving PER from 17.4% (our RNN–CTC) to 14.8%, and to 14.3% with multitask supervision (MFCC + BART). This result is robust across seeds ($\pm0.3\%$) and demonstrates a clear benefit of contextual, autoregressive decoding over framewise CTC when trained on the same neural signals and with matched front-ends. The performance gain is even more pronounced when compared to 19.7% PER from the original Willett et al. RNN–CTC formulation, indicating the impact of the introduced pre-encoder steps.

At the word level, our Transformer + MFCC + BART achieves 25.6% WER, improving upon the prior end-to-end baseline (BGRU-Phone from Feng et al., 26.3% WER). Adding the candidate generation and rescoring stage (see Appx. B for details) on top of the best multitask model further reduces WER to 19.4%, narrowing the gap to the two-stage hybrid system from Willett et al. (2023b) (17.4% WER via phoneme inference, n-gram hypothesis generation and external-LM rescoring) to 2 absolute points. The remaining performance gap is balanced by a notably faster inference rate, as described in 3.3, positioning our decoder as a competitive end-to-end alternative to the two-stage pipelines. Additional qualitative analysis of the word decoding is presented in Appx. A

Multitask objectives help. Adding the MFCC head improves PER across both RNN and Transformer families (e.g., from 14.8% to 14.6% PER for the Transformer). Adding BART lowers PER further (14.5%) and enables direct word decoding (26.0% WER), and the full MFCC + BART model yields the best numbers without rescoring (14.3% PER, 25.6% WER). We further investigated the impact of different BART initialization and optimization approaches (see Appx. C), presenting here the results of partially frozen decoder due to their balanced nature.

Replacing the proposed *Neural Hammer&Scalpel (NHS)* with either a global linear day transform (our implementation matching Willett et al. (2023b)) or no day transform degrades both PER and WER (Table 1; +2.7/3.6 PER points; +2.6/4.4 WER points). Combining a global affine "hammer" with a FiLM "scalpel" and a learned gate provides the flexibility needed to compensate for session non-stationarities beyond what global alignment can capture.

Beyond RNN–CTC and seq2seq architectures, two recent models provide additional context on the Willett et al. (2023a) intracortical dataset. Littlejohn et al. (2025) evaluate a streaming RNN-Transducer decoder on the same split but attempted-speech trials only, reporting substantially lower performance (39.1% PER, 57.9% WER), which underscores the difficulty of jointly learning neural–acoustic alignment and autoregressive prediction in this low-resource setting. A Transformer–CTC architecture was explored by Feghhi et al. (2025), achieving notable 5.68% WER on the competition split, showing the potential of custom Transformer definition and multi-step hybrid pipeline. However,

as only word-level error rates are reported and the evaluation split differs, we treat these results as qualitative architectural context rather than as directly comparable baselines.

## 3.1 SCALING LAWS

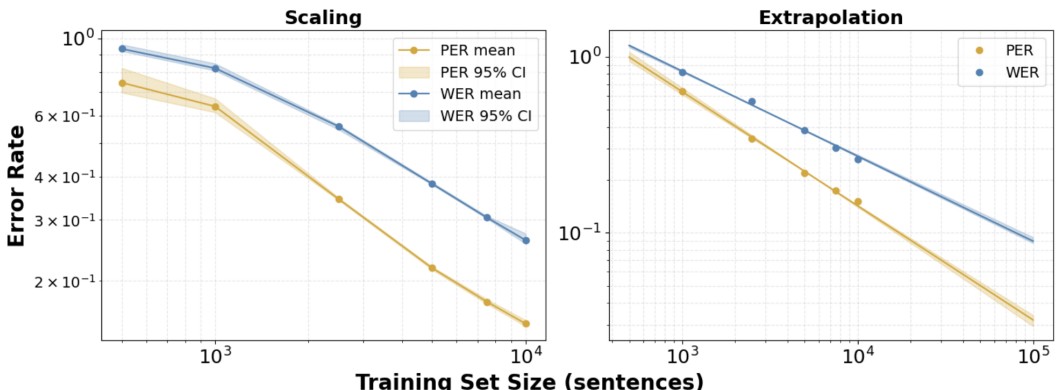

Figure 2: **Scaling behavior and power-law extrapolation Left:** Phoneme (PER) and word (WER) error rates for six dataset fractions $\{0.05, 0.10, 0.25, 0.50, 0.75, 1.00\}$, plotted as corresponding training-set sizes, with 95% bootstrap confidence intervals across seeds. **Right:** Power-law models fitted to the multi-seed mean curve (excluding the 0.05 fraction) with bootstrap 95% confidence bands and extrapolated performance up to 100,000 training trials.

We examine whether classical ML scaling laws—predictable power-law error reduction with more data—also emerge for intracortical speech decoding. Using day-stratified subsampling of the 8,800 training trials at fractions $\{0.05, 0.10, 0.25, 0.50, 0.75, 1.00\}$, we retrained our best-performing seq2seq variant (+MFCC+BART, no rescoring) with 5 seeds per fraction while keeping all training factors fixed. Figure 5 (left) shows PER and WER with bootstrap 95% confidence intervals ($N = 1000$). Seed variability is small, and both metrics follow a clean log–log linear trend consistent with scaling behavior reported in other domains (Kaplan et al., 2020). We then fitted a power law $e(N) = aN^b$ to the multi-seed mean curve and used bootstrap resampling to obtain confidence intervals for both parameters and predicted curves. Because the 0.05 fraction lies in a pre-asymptotic regime, we report extrapolations using fits that exclude this point (Figure 5, right). Under this model, expanding the dataset from 10k to 20k sentences is expected to reduce error from $14.3\% \rightarrow 8.4\%$ PER and $25.6\% \rightarrow 18.3\%$ WER, while a 100k-sentence dataset yields projected values of $\sim 2.9\%$ PER and $\sim 8.4\%$ WER.

These extrapolations should be interpreted with caution. Power-law fits assume that the neural data distribution remains sufficiently stationary as dataset grows, whereas intracortical signals can drift or change due to encapsulation effects, electrode degradation and recalibration. The confidence-bounded fits shown here should therefore be regarded as **optimistic lower-bound error estimates** achievable under stable recording conditions. Additional scaling curves for other model variants and confidence intervals are provided in Appx. D.

## 3.2 GENERALIZATION ON HELD-OUT DAYS

Real-world BCIs must cope with out-of-distribution neural data, including inference on days not seen during training. To test this, we trained our seq2seq Transformer (with MFCCs and a BART word decoder, no rescoring) either on all 24 recording days or on the first 21 days only. As a baseline, we evaluated the 24-day model on days 22–24 using day-specific transforms and report PER/WER deltas relative to this baseline (Fig. 3). We then reused the transform from the last training day (day 21) in two conditions: (i) the 24-day model re-evaluated on days 22–24 (*seen data + proximal*), and (ii) the 21-day model evaluated with the day-21 transform (*unseen data + proximal*).

On days included in training, the proximal transform changes PER/WER by at most $\sim$2 percentage points, indicating that NHS transforms for neighboring days are similar. For unseen days, the

normalized penalty starts around 6 percentage points and rises to roughly 10–13 percentage points as the temporal gap grows, revealing substantial non-stationarities in the neural signal. Together with Table 1, these results show that day transforms—and in particular the proposed NHS transform—substantially improve decoding, but in their current day-parametrized form they do not fully remove across-day distribution shifts; the remaining drift must be handled by the shared encoder–decoder model.

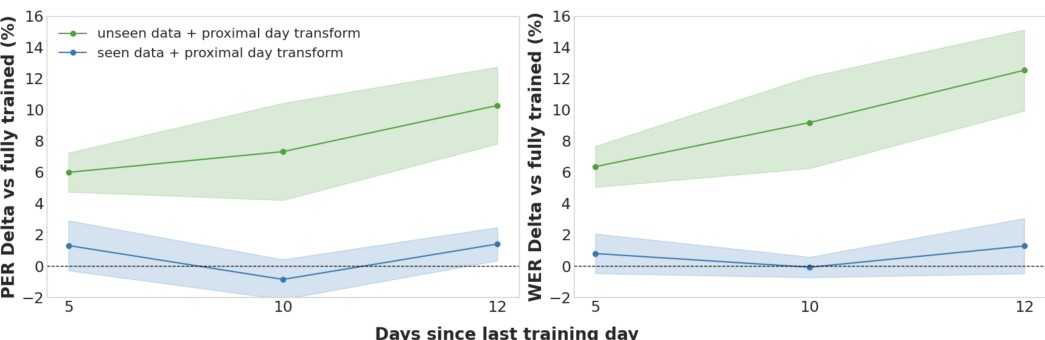

Figure 3: **Generalization.** PER/WER deltas on held-out days 22–24 relative to a 24-day model evaluated with day-specific transforms (0%). Blue curves (*seen data + proximal*) show the same model re-evaluated on these days using the transform from day 21. Green curves (*unseen data + proximal*) show a 21-day model evaluated on days 22–24 with the day-21 transform, after subtracting the mean 21- vs. 24-day performance gap on days 1–21. Mean over 5 seeds, shaded regions denote 95% bootstrap confidence intervals.

## 3.3 INFERENCE SPEED

We benchmarked wall-clock decoding on the test split (880 trials; 5,166 words total). All runs used a single GPU as indicated. Throughputs are reported as sentences/s and words/s; latencies are the mean time per sentence and per word.

Table 2: Inference time on the test set (880 sentences; 5,166 words). Times are wall-clock.

| Model | HW | Sent/s | Words/s | ms/sent | ms/word |
|---|---|---|---|---|---|
| Transformer + MFCC + BART | H100 | 19.13 | 112.30 | 52.3 | 8.9 |
| RNN–CTC + MFCC + BART | H100 | 30.34 | 178.14 | 33.0 | 5.6 |
| RNN–CTC + MFCC + Willett LM | A100 + CPU | 1.57 | 9.20 | 638.6 | 108.7 |
| Transformer + rescoring pipeline | H100 | 3.12 | 18.32 | 320.4 | 54.6 |

On a single H100, the purely end-to-end stacks decode the full test set in tens of seconds: Transformer+MFCC+BART completes in 46 s (19.1 sent/s; 112.3 words/s), while RNN–CTC+MFCC+BART finishes in 29 s (30.3 sent/s; 178.1 words/s). Adding the generation and rescoring stage on top of the Transformer increases runtime to 282 s (3.1 sent/s; 18.3 words/s), roughly $6\times$ slower than the plain Transformer but still about $2\times$ faster than the Willett-style RNN–CTC + WFST + LM pipeline. The latter configuration (our RNN–CTC + MFCC backbone with Willett LM) remains the slowest overall at 562 s (1.57 sent/s; 9.2 words/s), driven primarily by CPU-bound WFST search and large-LM rescoring. Although the hardware differs for the Willett LM configuration (A100 + CPU, with OPT rescoring incompatible with H100), these measurements reflect the practical decoding costs of each approach: hybrid pipelines achieve the lowest WERs but incur substantial wall-clock overhead from hypothesis generation and rescoring, whereas the lightweight end-to-end variants provide order-of-magnitude higher throughput on modern GPUs.

## 3.4 ATTENTION STRUCTURE

Our modeling requires aligning neural dynamics to phonemes and words without explicit temporal annotations of when speech units occur. The seq2seq Transformer, via attention, must therefore learn

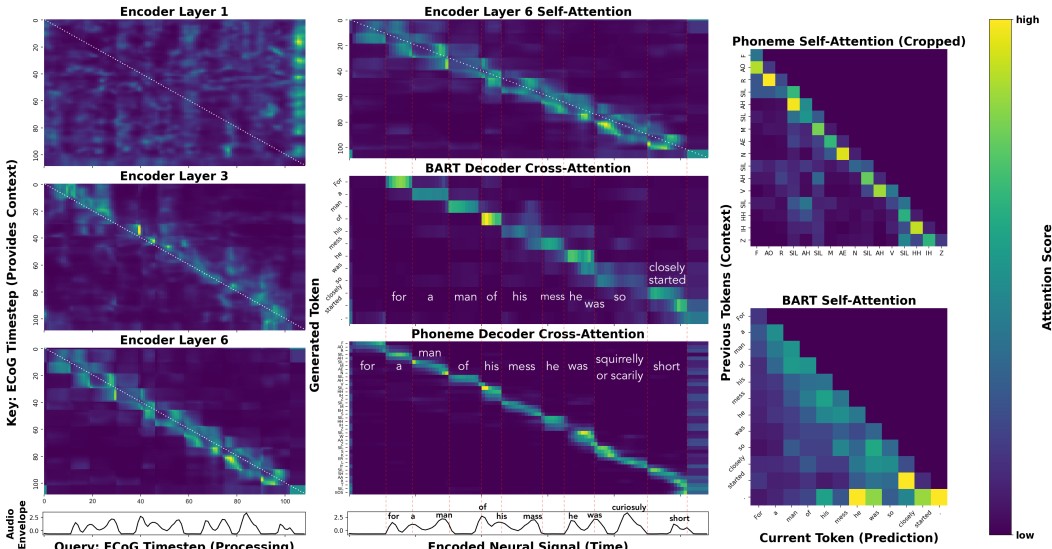

Figure 4: **Attention analysis.** Left: encoder self-attention for Layers 1, 3, and 6. Middle: cross-attention for BART and phoneme decoders. Right: decoder self-attention. Dotted vertical lines align with salient envelope events. Late encoder layers exhibit clean diagonals and "boxed" segments that decoders repeatedly attend to; phoneme self-attention is local, while BART self-attention is broadly triangular. The trial text is *"for a man of his mass he was curiously short"*, target words are approximately aligned to envelope features (no ground-truth timings exist for this dataset). For phoneme decoder the annotated word predictions were generated from phonemes using LLM

its own timing: the encoder organizes the input over time and the decoders retrieve relevant moments as tokens are generated. We analyze these mappings to visualize the model's emergent representations of neural activity in the context of speech decoding. Figure 4 shows typical attention patterns for a single, representative validation trial. In the encoder, early layers distribute attention broadly; by Layer 3, diagonals and compact local "blobs" emerge; by Layer 6, crisp diagonals interleave with boxed regions that partition the sequence into short segments whose boundaries qualitatively track onsets, offsets, or envelope peaks. Notably, the encoder aggregates information from both pre- and post-event timesteps, but within a limited span, suggesting that chunked/limited-lookahead streaming could preserve most offline behavior—at the cost of latency proportional to the lookahead window.

The decoders re-use these temporal chunks through cross-attention. The BART word decoder typically concentrates a word's evidence within a single segment (occasionally drawing on adjacent segments for cross-boundary words). The phoneme decoder resolves finer structure inside each segment: attention often begins with a strong, localized `<SIL>` (inter-word) focus, followed by a temporal cascade over singular word phonemes. We also note a mild temporal asymmetry—phoneme cross-attention tends to peak just before the envelope peak (consistent with preparatory neural activity), whereas word cross-attention can lag slightly (consistent with accumulating evidence). Both decoders sometimes assign mass to the beginning and end of the trial, which may serve as global anchors. Decoder self-attention mirrors their roles: the phoneme decoder is sharply local (n-gram-like), while BART maintains a broad, triangular pattern reflecting long-context language modeling from pretraining.

These observations are qualitative and drawn from a singular, albeit dataset-wise representative, example; We therefore interpret the figure as a plausible account of how the model often segments time and allocates evidence, rather than as a definitive mechanistic explanation. To provide more qualitative grounding to this analysis, the temporal chunking pattern was subjectively confirmed to appear in all 880 validation trials after visual inspection by all authors of this work. Additionally, in Appx. E the Figure 6 shows examples of the encoder-decoder temporal chunking emergence and usage across 20 randomly sampled trials and Figure 7 visualizes examples of this encoder-derived chunking across 4 model variants and 15 random trials. Although the experiments are mostly qualitative, the outlined patterns appear prevalent in model's working.

## 4 DISCUSSION

Seq2seq Transformers are a competitive choice for intracortical speech decoding in the low-data regime. On the Willett et al. (2023a) dataset our multitask variant sets a new PER benchmark, indicating that conditioning predictions on both full (unmasked) neural context and generated history helps when supervision is scarce. Attention analyses reinforce this picture: the encoder evolves from diffuse early layers to crisp, box-like segments that tile time, and both decoders reuse these segments via sharply localized cross-attention aligned with onsets/offsets. Such adaptive time-chunking and long-range integration are hard to realize with framewise CTC encoders that assume local, independent decisions. At the word level, the picture is more nuanced. With greedy decoding, our seq2seq model already improves on prior end-to-end baselines, but still lags behind two-stage hybrids in absolute WER. Extending the architecture with candidate generation and rescoring closes most of this gap: a simple linear combination of phoneme-level scores (PER and phoneme-head log-likelihood) and an external LM score reduces the WER gap to 2 absolute points while keeping decoding substantially faster. A complementary route would be to plug our phoneme predictions into the Willett's WFST+LM pipeline, where improved PER should translate into strong WER. In practice, we found that the released WFST expects framewise posteriors and is incompatible with our sequence-level logits; retraining of that pipeline is therefore left for future work.

Auxiliary tasks matter. MFCC supervision improves PER despite degraded audio, indicating that coarse acoustic cues provide a useful regularizer for early encoder layers. Adding a BART-based word head further nudges PER downward, indicating word-level supervision aiding phoneme representations. BART initialization experiments show that a randomly initialized decoder can nearly match pretrained variants, suggesting that in this data regime most useful linguistic structure is learned from the supervised task, with pretraining adding only modest gains. Day-to-day nonstationarity remains a key challenge for intracortical BCIs. The NHS transform consistently outperforms both a purely linear day transform and no transform on PER and WER, indicating that flexibility between global alignment and channel-wise calibration helps capture session-specific changes. Held-out-day experiments show that reusing a transform from a nearby day allows decoding on unseen days, but with a performance penalty (up to ∼10–13 percentage points) that grows with temporal distance. NHS therefore reduces across-day drift but, in its current per-day parameterization, is not sufficient for robust generalization, motivating time-relative or continuous session-conditioned calibration mechanisms that can better extrapolate to new days.

Scaling behavior is encouraging: experiments with day-stratified subsampling show that both PER and WER follow clean, approximately linear trends in log–log space, and across-seed variability is small relative to the overall scaling effect. Power-law extrapolation suggest that doubling the dataset to 20,000 sentences could more than halve both PER and WER, with projected single-digit error rates at 100,000 sentences. These projection are optimistic lower bounds: they assume approximate stationarity of the neural distribution and idealized asymptotic behavior, whereas real BCIs must contend with long-term drift, electrode degradation, and constraints on recalibration. From a deployment perspective, chunked or limited-lookahead encoders and stabilized emission policies could make the seq2seq stack streaming-compatible without fundamentally changing the recipe. Today's trial-level labels constrain supervision of phoneme/word timing; broader availability of timestamped annotations would support better training of emission policies and alignments. Ethical and safety considerations are central: decoding errors can misrepresent user intent, so systems should expose calibrated uncertainty, enable rapid correction, and provide safe fallbacks. Because attempted and inner speech may share representations (Kunz et al., 2025), intention-based operation (e.g., explicit "go" signals, robust "off" states) and protections against non-consensual decoding (Tang et al., 2023) are essential, alongside transparent opt-in/opt-out data handling.

This study has several limitations. Cross-participant or cross-implant generalization is not assessed. We focus on offline performance rather than fully streaming operation and do not yet evaluate continual adaptation over long timescales or under progressive electrode loss. Within these constraints, our contributions are primarily empirical and integrative: we show that multitask seq2seq Transformers can deliver state-of-the-art phoneme decoding, competitive and scalable word decoding with rescoring, interpretable internal structure via attention, and predictable scaling behavior on intracortical speech data. Promising directions include combining these models with stronger candidate-generation pipelines, extending NHS to continuous-time or multi-session settings, and leveraging self-supervised or cross-subject pretraining to move toward more general brain-to-text foundation models.

## ETHICS STATEMENT

All authors have read and will abide by the ICLR Code of Ethics. This study uses the publicly available intracortical dataset of Willett et al. (2023a); no new human data were collected. The original dataset was gathered under IRB oversight with informed consent, and all analyses were performed on de-identified recordings. We complied with the dataset license and did not attempt re-identification or linkage with external sources. Our models are intended to advance scientific understanding of neural speech decoding and are not suitable for clinical, legal, or surveillance use. Because attempted/inner speech may raise privacy concerns, we emphasize safeguards discussed in Section 4: intention-based operation (e.g., "go" signals, robust "off" states), calibrated uncertainty and user correction, conservative deployment practices, and opt-in/opt-out data handling. We disclose potential risks of misuse (misinterpretation of intent, unauthorized decoding) and recommend that any future deployment include explicit consent, human oversight, and protections against non-consensual use.

## REPRODUCIBILITY STATEMENT

We took multiple steps to ensure reproducibility. The dataset source and all preprocessing are documented in Section 2.1. The full model architecture, the NHS day transform, training objectives, training protocol, and evaluation metrics are specified in Section 2.2 and Section 2.3. Hyperparameters are listed in Appendix F. The scaling-law protocol (fractions and fitting) appears in Section 3.1. Attention extraction procedures are described in Appendix E. We report means and standard deviations over five seeds in Table 1 and provide exact splits (stratified by day) in our code release. An anonymized code package containing training/inference scripts and configuration files is provided in the supplementary materials.

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

## A  QUALITATIVE ERROR ANALYSIS

Table 3: Example word-level predictions of Transformer+MFCC+BART model (no rescoring) from 10th, 50th (median) and 90th WER percentiles.

| WER | %ile | Target sentence | Predicted text |
|---|---|---|---|
| 0.0 | 0 | Do you know where it might have gone? | Do you know where it might have gone. |
| 0.0 | 0 | I am an artist, lost in my own vision. | I am an artist, lost in my own vision. |
| 0.0 | 0 | Read the decision below. | Read the decision below. |
| 0.2 | 50 | I don't think so anymore. | I don't think so many. |
| 0.2 | 50 | Just **way** in the back. | Just **why** in the back? |
| 0.2 | 50 | Sometimes they're not very **open**. | Sometimes they're not very **hard**. |
| 0.6 | 90 | **Fifty** nine **kilometers per** gallon. | **Fifteen** nine **automobiles for** gallon. |
| 0.6 | 90 | We were **promised civil liberties**. | We were **most several places**. |
| 0.6 | 90 | **Special rules** for employment **cases**. | **Paces** for employment **taxes**. |

Table 3 shows representative predictions of Transformer + MFCC + BART model (no rescoring) at the 10th, 50th (median), and 90th WER percentiles. Errors at the median and tail are dominated by phonetic confusions with high acoustic similarity (e.g., *way→why*, *fifty→fifteen*; *cases→taxes*), often preserving sentence rhythm but shifting semantics. This suggests that the neural signal supplies strong sub-lexical acoustic evidence while semantic disambiguation remains limited without explicit rescoring.

## B  GENERATION STRATEGIES AND SCORING ANALYSIS

We employed two complementary decoding configurations to construct a rich hypothesis space for the rescoring stage:

- **Beam-20 (deterministic).** A standard beam search with `num_beams=20`, producing 20 high-likelihood candidates per trial. This method explores the most probable regions of the model's output distribution in a deterministic manner.

- **Nucleus-128 (stochastic).** A high-diversity sampling setup that generates 128 candidates using nucleus sampling (`top_p=0.95`, `temperature=1.0`, `top_k=0`). Unlike beam search, this strategy introduces stochasticity and explores a broader portion of the distribution, often producing phrasings that would not appear under purely probability-maximizing generation.

Table 4 summarizes the effect of different candidate selection approaches. The default greedy decoding provides a strong baseline, while beam search offers only a modest improvement when simply the highest overall-probability beam output is chosen. More substantial gains arise when candidates are rescored using phoneme-level evidence (PER-score and phoneme-score), especially compared to LLM-only scoring–highlighting the weakness of BART decoder to differentiate in the acoustic space as shown in A. The best overall performance is obtained when all scoring components are combined with ratio PhonemeHead:PER:LLM 9:4:5, lowering the final WER from 25.6% to

Table 4: Word-level performance comparison between strategies of candidate selection. Score-based results are obtained by selection of candidates generated using beam search and stochastic sampling strategies. All results reported as mean ± SD across 5 global seeds

| Model | WER (%) |
|---|---|
| Transformer Seq2Seq + MFCC + BART | |
| Greedy decoding | $25.6 \pm 0.2$ |
| Beam search (top result) | $24.8 \pm 0.4$ |
| PER score only | $23.2 \pm 0.4$ |
| Phoneme head score only | $22.6 \pm 0.2$ |
| LLM score only | $27.2 \pm 0.4$ |
| All scores combined | $19.4 \pm 0.3$ |
| Oracle | $14.5 \pm 0.2$ |

19.4% PER (delta 6.2 pp.). The word-level performance could theoretically be improved further as the oracle (always choose the best) WER is 14.5%.

## C  BART INITIALIZATION AND TRAINING: LINGUISTIC PRIOR IMPACT

We examined whether word-level linguistic priors encoded in the pretrained BART decoder influence decoding performance. In Stage 2, we compared three configurations: (i) initializing the BART decoder from the `bart-base` checkpoint while freezing its first three layers to preserve pretrained linguistic structure, (ii) initializing from `bart-base` and finetuning all layers, and (iii) replacing the decoder with an identical BART architecture initialized with random weights. In all conditions, the same pretrained BART tokenizer was used, so the subword vocabulary and segmentation were held fixed.

As shown in Table 5, all strategies achieve highly similar phoneme and word error rates. The fully finetuned pretrained model yields the best WER, but the difference relative to random initialization is small. Notably, the randomly initialized decoder performs on par with the pretrained variants, despite lacking any large-corpus language modeling priors in its weights. This suggests that the linguistic structure exploited during decoding arises primarily from patterns in the supervised dataset and from the encoder's neural representations, rather than from the pretrained BART language model itself.

Comparison to Table 1 shows that adding the word-level BART decoder—regardless of how its weights are initialized—reduces phoneme error rates relative to a phoneme-only baseline. This indicates that the encoder representations are shaped by word-level supervision even when the decoder itself carries no pretrained linguistic knowledge. Taken together, these results imply that our transformer encoder can learn higher-level, linguistically structured representations directly from neural activity and task data, with pretrained BART weights providing at most a modest additional benefit in this regime.

Table 5: Performance comparison between different strategies of BART initialization and training. Results reported as mean ± SD across 5 global seeds

| Model | PER (%) | WER (%) |
|---|---|---|
| Transformer Seq2Seq + MFCC + BART | | |
| BART pretrained + Freeze first 3 layers | $14.3 \pm 0.3$ | $25.6 \pm 0.2$ |
| BART pretrained + No freezing | $14.6 \pm 0.4$ | $\mathbf{25.2 \pm 0.1}$ |
| BART random initialization | $\mathbf{14.2 \pm 0.2}$ | $25.8 \pm 0.4$ |

# D SCALING AND EXTRAPOLATION

Figure 5 shows near-linear trends in log–log space for both PER and WER, consistent with power-law scaling observed in other domains (Kaplan et al., 2020). Auxiliary supervision shifts the curves downward at every fraction, indicating improved data efficiency.

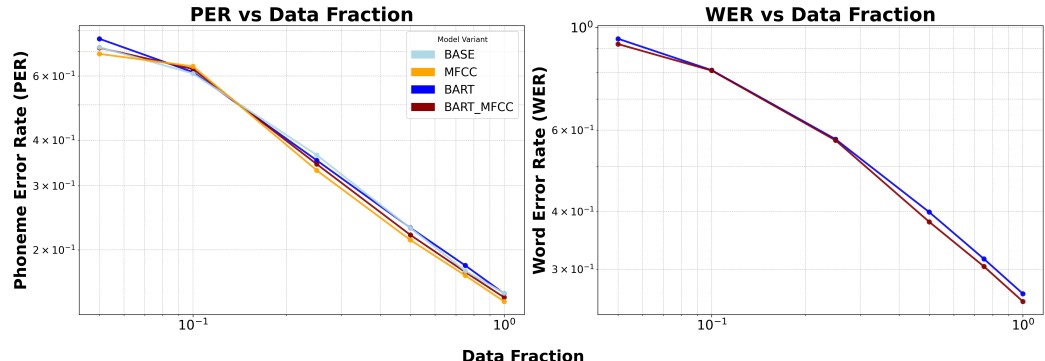

Figure 5: **Scaling behavior.** PER (left) and WER (right) versus the fraction of training trials used ({0.05, 0.10, 0.25, 0.50, 0.75, 1.00}) on log–log axes. Points are single-seed runs; only training-set size is varied (all other settings held fixed). All variants follow clear power law scaling; auxiliary supervision (MFCC, BART) improves data efficiency across fractions.

Table 6: Extrapolation of performance with increased train set size (fraction). Results reported as mean across 5 seeds with bootstrapped across-seeds 95% confidence interval when fitting power-law coefficients. Extrapolation values provided in scenarios of excluding the first, pre-asymptotic, fraction and all fractions

| Train-set size | PER (%) | WER (%) |
|---|---|---|
| 8,800 (current) | 14.3 (14.1, 14.5) | 25.6 (24.9, 26.1) |
| Excluding pre-asymptotic fraction | | |
|    20,000 (2x) | 8.4 (8.0, 8.6) | 18.3 (17.6, 18.9) |
|    100,000 (10x) | 2.9 (2.7, 3.1) | 8.4 (7.8, 8.9) |
| Including all fractions | | |
|    20,000 (2x) | 11.0 (10.5, 11.4) | 21.5 (21.0, 22.0) |
|    100,000 (10x) | 3.8 (4.3, 5.1) | 11.35 (10.8, 11.7) |

# E ATTENTION EXTRACTION AND FURTHER EXAMPLES

To analyze model behavior, we extracted attention maps from the encoder and both decoders. The PyTorch TransformerEncoder and phoneme TransformerDecoder were modified setting nn.MultiheadAttention class attribute `return_weights=True` to return attention weights for each layer. For the word decoder, we used the Hugging Face BART implementation with `output_attentions=True` to visualize attention heads across tokens. These attention maps allow us to interpret which neural timesteps contribute to specific phoneme or word predictions, shedding light on model alignment and representation dynamics. Figures 6 and 7 provide further examples of the observed patterns among randomly sampled 20 and 15 validation trials respectively.

# F TRANSFORMER MODEL PARAMETERS

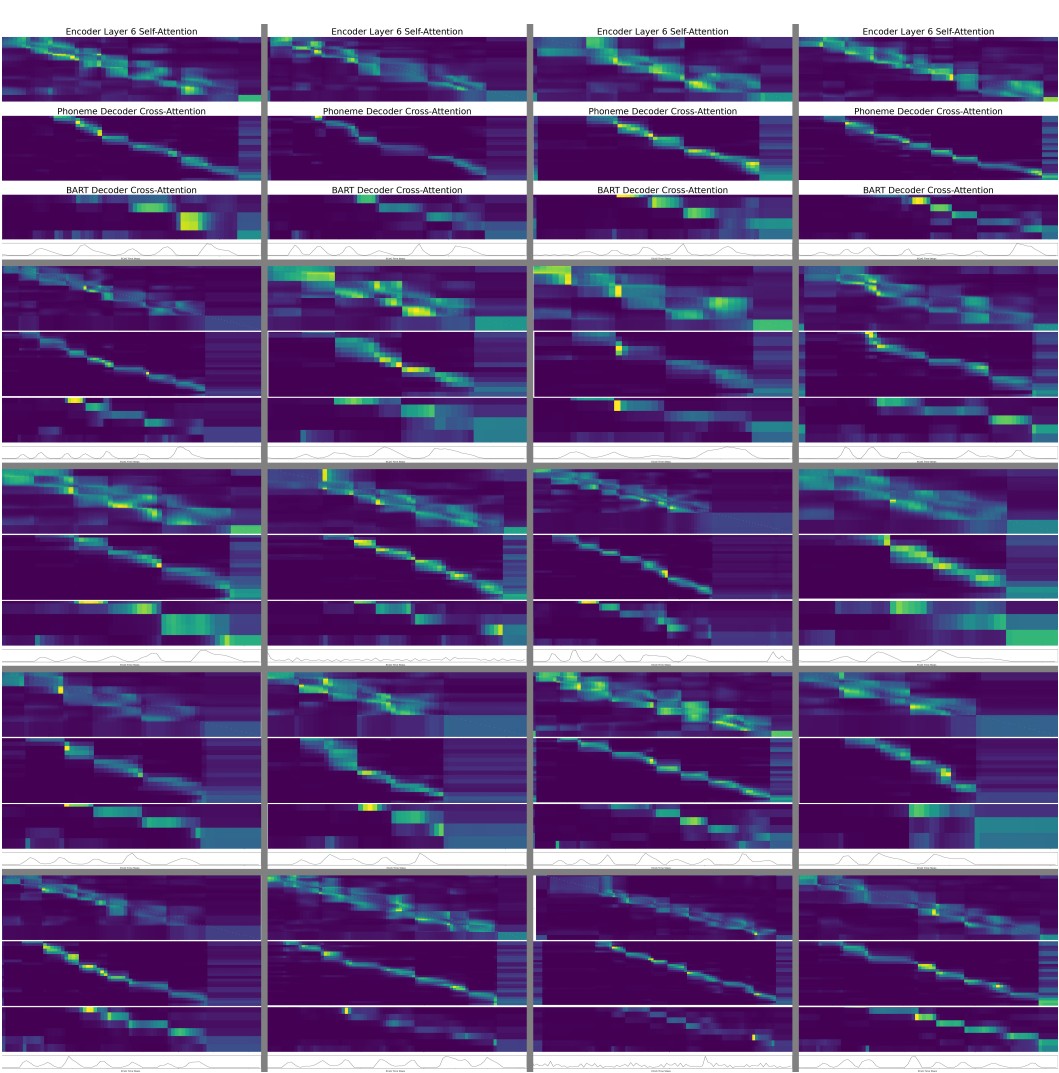

Figure 6: **Self and cross attention: Usage of temporal chunking.** Each cell shows vertically stacked 1) Encoder self-attention in the last layer, 2) Phoneme Decoder cross-attention, 3) BART word decoder cross-attention, 4) Audio envelope, obtained from Transformer Seq2Seq + MFCC + BART model, visualized for 20 randomly chosen validation trials. The pattern of temporal chunking of neural signal in the encoder and the downstream usage of those chunks by both decoders is prevalent across all samples - as well as the temporal lag of word decoder relative to the phoneme decoder, indicating evidence accumulation.

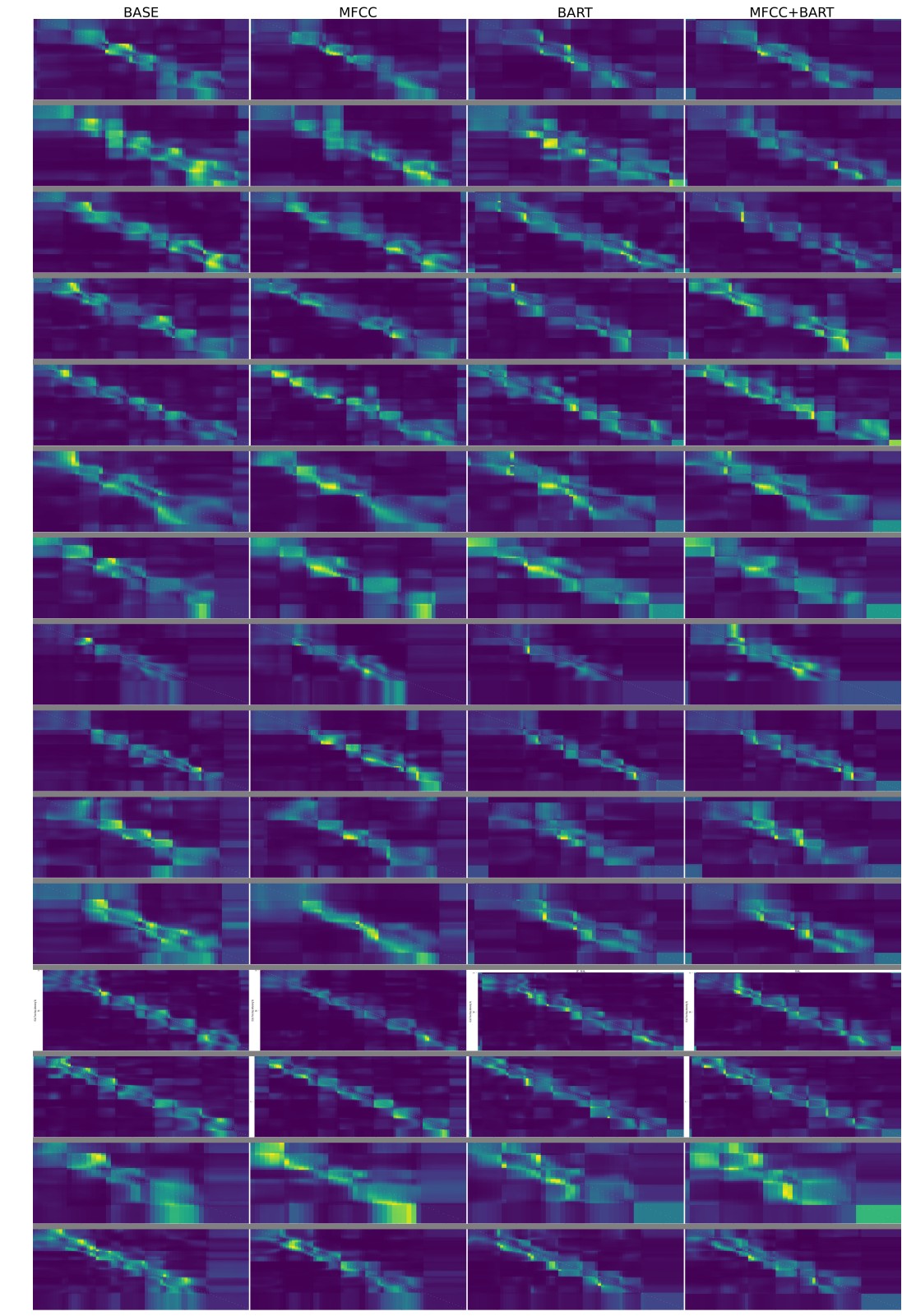

Figure 7: **Self-attention: Temporal chunking.** Self-attention map of the last (6th) layer of encoder across model variants (columns) visualized for random 15 trials (rows). The representation pattern of temporal chunking of the neural signal into "boxes" is prevalent across the trials regardless of the auxiliary objectives

Table 7: Training and model parameters for the two-stage training process of Transformer+MFCC+BART model. Stage 1 focuses on pre-training the ECoG encoder with an auxiliary MFCC prediction task alongside the main phoneme decoding task. Stage 2 introduces a BART-based word decoder head and fine-tunes the entire model jointly on phoneme and text decoding. The Stage 2 column only lists values in Multi-Task Setup section, as only then they differ from Stage 1.

| Parameter | Stage 1 Value | Stage 2 Value |
|---|---|---|
| GPU | Nvidia H100 80GB of HBM3 memory | |
| **Core Transformer Architecture** | | |
| Model dimension (`d_model`) | 512 | |
| Attention heads (`n_head`) | 8 | |
| Encoder layers | 6 | |
| Decoder layers | 6 | |
| Feed-forward dimension | 2048 | |
| Activation function | GELU | |
| **Input Processing & Feature Extraction** | | |
| ECoG features | 256 (128 channels x 2 features per-channel) | |
| Feature extractor | Binned Attention Conv | |
| Conv kernel size | 5 | |
| Downsampling strategy | Convolutional | |
| Downsampling factor | 4 | |
| Day adaptation | NHS | |
| Number of days | 24 (all) | |
| **Regularization & Augmentation** | | |
| Dropout | 0.4 | |
| Time masking probability | 0.3 | |
| Max time mask length | 25 steps | |
| Max time mask proportion | 20% | |
| Channel masking probability | 0.25 | |
| Max channels masked | 15 electrodes | |
| **Multi-Task Setup** | | |
| Training objective | Joint phoneme decoding & MFCC prediction | Joint phoneme & text decoding |
| Phoneme decoder loss weight | 1.0 | 1.0 |
| MFCC auxiliary task | Enabled | Disabled |
| MFCC head type | Linear | N/A |
| MFCC aux. loss weight | 0.001 | N/A |
| Aux. head input layer index | 1 | N/A |
| BART text decoder task | Disabled | Enabled |
| BART model type | N/A | `bart-base` |
| BART loss weight | N/A | 1.0 |
| BART freezing strategy | N/A | Freeze first 3 decoder layers |
| **Optimizer & Scheduler** | | |
| Optimizer | AdamW | |
| Learning rate | 1e-4 | |
| Weight decay | 1e-3 | |
| Scheduler | ReduceLROnPlateau | |
| Scheduler metric | `val_per_sg_epoch` | |
| Scheduler factor | 0.5 | |
| Scheduler patience | 10 epochs | |
| **Training Details** | | |
| Batch size | 4 | |
| Gradient accumulation | 5 steps | |
| Effective batch size | 20 | |
| Epochs | 200 | |
| Precision | 32-bit float | |
| ECoG Encoder frozen | No | |
| Phoneme Decoder frozen | No | |

