# OpenReview forum: "Don’t Forget the Context: A Multitask Transformer for Intracortical Speech Decoding"
_ICLR.cc/2026/Conference — Submitted to ICLR 2026_

### Official Review · Reviewer_1HWu · 2025-10-29

**Soundness:** 2
**Presentation:** 1
**Contribution:** 2
**Rating:** 4
**Confidence:** 4

**Summary:**

The authors present a sequence-to-sequence (seq2seq) Transformer model for decoding open-vocabulary speech from intracortical neural signals. To address data scarcity, the model is trained in a multitask framework that includes phoneme decoding, word decoding (using a pretrained BART head), and an auxiliary regression task on MFCCs. The paper also introduces a novel day-specific transform, the "Neural Hammer & Scalpel" (NHS), to mitigate cross-session non-stationarity. The authors claim their model sets a new state-of-the-art in phoneme decoding and improves upon a previous end-to-end baseline for word decoding.

**Strengths:**

- The paper provides a strong rationale for moving beyond RNN+CTC models, clearly outlining their limitations, such as the conditional-independence assumption and the separation of the neural encoder from the language model.
- The model claims to achieve a new state-of-the-art (SOTA) Phoneme Error Rate (PER) of 14.3%. This demonstrates that the seq2seq architecture and multitask setup are effective for learning robust phoneme-level representations.
- The scaling analysis (Fig. 2) provides good evidence that the model architecture can improve with more data. The attention visualizations (Fig. 3) offer valuable qualitative insights into the model's learned alignments.

**Weaknesses:**

- The primary weakness is the model's performance on the main task. The paper's best-reported WER is 25.6%. This is significantly worse than the established 17.8% WER from the hybrid RNN-CTC + LM baseline (Willett et al., 2023b) reported in the same table. An absolute performance gap of 7.8% on the primary metric is too large to overlook, especially for a clinically-motivated application.
- The paper's methodological contributions are largely combinations of existing techniques.
  - The use of seq2seq Transformers is a standard, established practice in automatic speech recognition (ASR). Applying it to neural signals is a logical, but incremental, step.
  - The "Neural Hammer & Scalpel" (NHS) transform is a novel combination of a per-day affine transform (which the authors note is similar to a prior baseline) and a FiLM-style modulation. This is a good engineering contribution but not a fundamental new method for adaptation.
  - Multitask learning is a common regularization technique.
- The paper claims to improve "over previous end-to-end systems in word decoding" by comparing its 25.6% WER to the 26.3% WER of Feng et al. (2024). While true, this comparison obscures the fact that both end-to-end systems perform substantially worse than the existing, simpler hybrid-model baseline (17.8% WER).

In my opinion, the paper's strongest result is its SOTA phoneme decoding. The work might be better received if it were reframed to focus on this achievement. By framing the paper as a word decoder (as in the title and abstract), it invites a direct comparison to the SOTA word decoder, a comparison it does not win. Focusing on the value of seq2seq and multitask learning for phoneme-level representation learning would be a more defensible claim.

**Questions:**

- The authors have shown the model is a superior phoneme decoder (14.3% PER vs. 17.4% for the RNN-CTC). What happens if the authors use their Transformer model as just a phoneme generator and feed its output into the same WFST + LM rescoring pipeline used by the Willett et al. (17.8% WER) baseline? This would provide a direct, apples-to-apples comparison of the neural encoder quality and test if the improved PER can actually lead to a better WER.
- NHS transform introduces 24 separate sets of "hammer" and "scalpel" parameters for the 24 days. How does this model generalize to a hypothetical 25th day not seen in the training set? Does this per-day parameterization risk overfitting and limit the model's ability to generalize to new, unseen sessions?

---

> ### Author Response · Authors · 2025-11-28
>
> 1. Word-level performance vs. hybrid WFST+LM baseline.
>
> We agree that the original submission did not sufficiently address the large WER gap to the Willett RNN–CTC + WFST + LM pipeline, and we have clarified our positioning in the revised manuscript.
> First, in response to this and other reviews we implemented the reviewer’s suggestion to exploit our model’s fast inference and add a second pass. In the revised paper we now: 1) generate multiple sentence hypotheses per trial using a mix of beam search and stochastic sampling, and 2) Rescore them using a linear combination of (i) PER-based scores, (ii) phoneme-head log-likelihood, and (iii) an external LM score.
>
> This generation+rescoring stage (Sec. 2.2 and Appx. B) reduces WER from 25.6% (greedy) to 19.4%, bringing our seq2seq system to within 2 absolute WER points of the Willett hybrid baseline (17.4%) while remaining substantially faster at inference (Table 2). In Results and Discussion sections we clearly state the WER-wise superiority of the two-stage hybrid systems; the comparison to previous end-to-end work provides additional context to our modelling approach - a lightweight end-to-end alternative with a favorable accuracy–latency tradeoff.
>
> 2. Feeding the phoneme outputs into the WFST + LM pipeline (Q1).
>
> We fully agree that this would be an ideal “apples-to-apples” test of whether improved PER translates into better WER under the same WFST+LM setup. We attempted this in preliminary work but ran into a nontrivial compatibility issue as the released Willett WFST pipeline expects framewise phoneme posteriors aligned to fixed 20 ms frames (RNN–CTC outputs), while our seq2seq phoneme head is autoregressive and produces a probability distribution over token positions, not over every time frame. Bridging this gap would require retraining the WFST (or training an additional framewise phoneme predictor) under our preprocessing and alignment scheme, which we could not do within the rebuttal window. We now state this limitation explicitly in the paper and list WFST retraining with our sequence-level logits as a concrete direction for future work. Conceptually, we agree this is a very informative experiment and intend to pursue it.
>
> 3. NHS per-day parameterization and generalization to unseen sessions (Q2).
>
> We share the concern that per-day parameters could overfit and limit generalization. In the revision we added exactly the kind of test the reviewer suggests: a held-out-day generalization experiment (Sec. 3.2, Fig. 3). We train our best seq2seq model on either all 24 days or only days 1–21. We then evaluate on days 22–24 under three conditions: 1) 24-day model with its own day-specific transforms (baseline), 2) 24-day model with the day-21 transform reused on days 22–24 (“seen data + proximal”), 3) 21-day model evaluated on days 22–24 with the day-21 transform (“unseen data + proximal”), after normalizing for the overall 21 vs 24-day performance gap.
>
> Results show that reusing a transform from a temporally proximal day changes PER/WER by at most ~2 percentage points on days that were seen during training, but for truly unseen days, penalties grow to ~6–13 percentage points as the temporal gap increases. Together with the NHS vs Linear vs No-transform ablations in Table 1, this suggests that NHS clearly improves performance relative to linear or no day transform, however in its current per-day parametrization it also does not fully solve generalization to new sessions; the encoder–decoder still carries much of the burden, and drift over longer gaps remains visible.
>
> In the Discussion we now frame NHS as a first step, and propose continuous-time or session-conditioned NHS (e.g., generating parameters as a smooth function of days since implantation or a session embedding) as a key next step to handle a hypothetical “25th day” and beyond without retraining.
>
> 4. Incremental vs fundamental methodology
> We agree that our work is not an algorithmic breakthrough in the sense of introducing a fundamentally new model class. At the same time, we believe that (i) proposing and validating a seq2seq alternative to framewise CTC for intracortical speech, (ii) systematically analyzing multitask supervision and day-wise calibration, and (iii) providing scaling and interpretability insights on a public dataset constitute a useful methodological and conceptual contribution, in line with ICLR’s interest in important application areas and interpretability of learned representations.

---

### Official Review · Reviewer_Sx4q · 2025-10-29

**Soundness:** 3
**Presentation:** 2
**Contribution:** 2
**Rating:** 4
**Confidence:** 3

**Summary:**

This paper proposes a multitask seq2seq Transformer for intracortical speech decoding, directly mapping neural activity to phoneme and word sequences. It introduces Neural Hammer & Scalpel (NHS), a day-specific calibration module that mitigates cross-day drift, and adopts multitask training with auxiliary MFCC prediction to improve data efficiency. Experiments on the Willett et al. dataset show state-of-the-art phoneme decoding and competitive word-level results. Analyses of attention patterns and scaling laws further reveal interpretable temporal structure and consistent data–performance trends. Overall, the paper demonstrates that Transformer-based decoding can effectively capture contextual and linguistic information from intracortical recordings.

**Strengths:**

(1) Originality
- The paper presents a meaningful step forward in intracortical speech decoding by introducing a unified seq2seq Transformer architecture that jointly models neural and linguistic dynamics. While Transformers themselves are not new, their application in this specific domain, particularly with multitask supervision and explicit day-wise calibration—is novel and well-motivated. The proposed Neural Hammer & Scalpel (NHS) module is an way to address day-to-day nonstationarity, a major challenge often glossed over in prior work.

(2) Quality
- The experiments are carefully executed, with fair baselines and ablations that isolate the impact of each design component (e.g., NHS, MFCC supervision, BART conditioning). The inclusion of scaling law analysis and attention visualization adds depth and scientific rigor beyond simple performance reporting. The model achieves state-of-the-art phoneme decoding accuracy and competitive word-level results despite limited data.

(3) Clarity:
- The paper is generally clear and well-structured. Each architectural choice is motivated by a concrete empirical or neuroscientific problem (e.g., temporal drift, weak audio supervision). Figures and tables effectively support the arguments, and the explanations of the day-adaptation mechanism are particularly easy to follow.

(4) Significance:
- This work bridges a methodological gap between modern sequence modeling and practical neural prosthetic applications. By showing that Transformer-based decoding can capture contextual and linguistic dependencies directly from intracortical activity, the paper paves the way for future foundation-style “brain-to-text” models. It provides both a technical contribution and a broader conceptual shift for the speech BCI field.

**Weaknesses:**

(1) Limited novelty in model design
- Most components like Transformer backbone, multitask setup, FiLM modulation are adaptations of existing techniques rather than fundamentally new inventions. The originality lies mainly in integration and application. The paper could be strengthened by articulating why this particular combination works better than other possible architectures (e.g., conformer-based, latent-alignment models).

(2) Insufficient evaluation diversity
- Experiments focus heavily on a single dataset (Willett et al.), with no cross-subject or cross-task validation. This limits claims of generalization and practical robustness. A small-scale transfer or held-out-day test would better demonstrate NHS’s effectiveness beyond memorizing per-day patterns.

(3) Ablation depth and analysis scope
- While ablations exist, some design choices (e.g., the gating function or MFCC weight) lack sensitivity analysis. It would help to quantify how each auxiliary loss contributes to performance and stability over time. Similarly, results are mostly quantitative; additional qualitative error analyses (e.g., semantic vs. phonetic errors) would provide richer insight.

**Questions:**

(1) The proposed NHS module effectively addresses day-to-day nonstationarity, but it appears to rely on per-day embeddings learned jointly with the training set.
- Have you tested the model’s ability to generalize to unseen days (e.g., a held-out-day split)?
- If not, could you comment on whether NHS can handle new sessions without retraining, or how a continuous-time version might perform?

(2) The paper shows that adding MFCC and BART supervision improves performance, but the mechanism remains somewhat unclear.
- Do you have any analysis (e.g., layer probing, representation similarity) indicating how MFCC or BART signals influence encoder representations?
- Would the same benefit persist if MFCCs were randomly shuffled or misaligned?

(3) The scaling law results are interesting but extrapolated from relatively small data fractions.
- How sensitive are these fits to the chosen data fractions (0.1–1.0)?
- Have you validated that the power-law trend holds when adding or removing entire recording days rather than random subsets?

(4) The model uses a BART decoder with partially frozen layers.
- Did you explore alternative strategies, such as training from scratch or using a smaller LM head?
- How crucial is the BART initialization compared to a randomly initialized decoder for achieving good WER?

Overall, the experimental setup is comprehensive, with ablations and analyses. However, the work feels more like a domain-focused integration study rather than a conceptual or algorithmic innovation typical of ICLR. Its strength lies in methodological rigor and neuroscientific relevance, which might make it a better fit for a specialized journal.

---

> ### Author Response · Authors · 2025-11-28
>
> We thank the reviewer for the detailed and constructive assessment.
> (1) Limited novelty in model design.
>  We agree that the individual components of our architecture (Transformer backbone, multitask setup, FiLM-style modulation) are not algorithmically novel in general-purpose ML. Our goal is instead to provide a conceptual and empirical advance for intracortical speech decoding by (a) proposing a seq2seq alternative to the dominant framewise CTC paradigm, and (b) showing how multitask supervision and explicit day-wise calibration interact in this domain.
> CTC-based decoders, which are standard in speech BCIs, handle missing temporal alignment by making local, conditionally independent framewise decisions, later combined with a separate LM. Our seq2seq model removes both assumptions: the decoder conditions on the full neural context and its own output history, and we show that this leads to a substantial PER improvement over RNN–CTC baselines trained on the same features and day transforms (Table 1). This provides an alternative “recipe” for handling the alignment problem that we believe is conceptually important for the field, even if each ingredient is familiar.
> We fully agree that other modern architectures (e.g., conformers, latent-alignment models) are promising candidates. In early development we tested a Conformer encoder (achieving similar performance to the Transformer, though not fully optimized), but for the main study we chose not to systematically compare backbones, primarily for compute reasons and to keep the work focused on a single architecture while varying supervision and day adaptation.
> (2) Evaluation diversity and generalization (Weakness 2, Q1).
> In the current work we deliberately focus on the public Willett et al. dataset to allow direct comparison with existing work and detailed analysis of a single, well-characterized system.
> To address the concern about NHS generalization now includes a held-out-day experiment (Section 3.2, Fig. 3). We train our best seq2seq variant on (a) all 24 days and (b) only days 1–21, and then:
> 1) Evaluate the 24-day model on days 22–24 using their own day-specific transforms (baseline).
> 2) Reuse the day-21 transform on days 22–24 for the 24-day model (“seen data + proximal”).
> 3) Evaluate the 21-day model on days 22–24 with the day-21 transform (“unseen data + proximal”), after normalizing out the overall 21- vs. 24-day performance gap.
>
> We find that reusing a proximal transform changes PER/WER by at most ~2 percentage points on days seen during training, but leads to penalties of ~6–13 percentage points on truly unseen days as the temporal gap grows. Combined with the day-transform ablations in Table 1, this indicates that NHS does meaningfully reduce across-day drift but, in its current per-day parametrization, does not solve generalization to arbitrarily new sessions.
>
> (3) Ablation depth, auxiliary losses, and qualitative errors (Weakness 3, Q2).
>  We agree that deeper sensitivity analysis of some design choices would be valuable. In practice, MFCC and BART supervision were introduced because they consistently improved PER in pilot experiments, and we chose a conservative MFCC weight to treat it as a weak regularizer rather than a strong target (also scaled relative to the absolute loss magnitude).
> To provide more word-level insight beyond aggregate metrics, we now include a qualitative error analysis (Appendix A, Table 3): examples at different WER percentiles show that errors are dominated by phonetic confusions (e.g., way → why, fifty → fifteen), often preserving acoustic/phonetic structure while shifting semantics.
> (4) Scaling laws (Q3).
>  In the revised manuscript we strengthened the scaling analysis. We re-ran the scaling experiments for our best seq2seq variant (+MFCC+BART) with 5 seeds per dataset fraction and now show bootstrap 95% confidence intervals for PER and WER, as well as confidence intervals over fitted power-law parameters and extrapolated curves (Fig. 2 and Appendix D).
> We decided not to run additional experiments that remove entire recording days, because this would require modifying the validation set as well and would complicate comparison across conditions. Instead, robustness to unseen days is examined explicitly in the new Section 3.2.
>
> (5) BART strategies and importance of initialization (Q4).
> As requested, we explored several BART decoder variants (Appendix C):
> 1) Pretrained BART with the first 3 layers frozen (our main setting).
> 2) Pretrained BART with all decoder layers fine-tuned.
> 3) The same architecture with random initialization.
>
> All three achieved very similar PER and WER. The fully fine-tuned pretrained model gave slightly better WER, while the randomly initialized decoder nearly matched it and in fact achieved the best PER. This suggests that, in this low-data regime, most useful linguistic structure is learned from the supervised neural-to-text task itself, and that BART initialization is helpful but not critical.

---

### Official Review · Reviewer_cjht · 2025-11-02

**Soundness:** 3
**Presentation:** 3
**Contribution:** 3
**Rating:** 6
**Confidence:** 4

**Summary:**

This paper presents a transformer-based sequence-to-sequence model for decoding speech from intracortical neural recordings in a patient with ALS and anarthria. The authors propose a multitask framework that jointly learns phoneme and word decoding with auxiliary MFCC supervision, and introduce a "Neural Hammer & Scalpel" (NHS) day-specific transformation to handle cross-day nonstationarity. the model achieves 14.3% phoneme error rate (PER) and 25.6% word error rate (WER), improving over previous end-to-end approaches. The authors demonstrate favorable power-law scaling trends and provide attention visualizations showing interpretable temporal chunking aligned with speech structure.

**Strengths:**

1. Strong ablation studies, Table 1 systematically evaluates architectural choices (Transformer vs. RNN), auxiliary tasks (MFCC, BART), and day transformations (NHS vs. Linear vs. None)
2. Strong phoneme performance and clear improvements over framewise CTC approaches
3. Excellent presentation and contextualization
4. Good details for facilitating reproducibility

**Weaknesses:**

1. Scaling law extrapolations (Section 3.4) are based on single seeds; quantitative uncertainty is lacking.
2. The paper’s power-law scaling analysis (Section 3.4; Appendix B; Figure 2) extrapolates phoneme and word error rates from ∼10 k to ∼100 k training trials under the assumption that the neural data distribution remains stationary. This assumption is unlikely to hold for intracortical recordings. Therefore, the statement of the projected “low single-digit PER and WER” estimates (page 6, line 319) may be overly optimistic.
3. While Figure 3 presents intriguing qualitative patterns, the attention interpretation (Section 3.6) is based on single representative trial; needs statistical validation across dataset. The authors should quantify: (a) what fraction of trials exhibit clean "box" structure in Layer 6, (b) how cross-attention entropy varies across decoder types, (c) whether attention peaks align with envelope features statistically across the test set
4. The authors state "training stability and performance to be highly sensitive to hyperparameters" but provide no ablation over critical choices like model dimension, number of layers, dropout rate, or warmup schedule.

**Questions:**

1. Can you provide confidence intervals or multiple-seed averages for the power-law fits in Figure 2? How sensitive are the extrapolations to the chosen functional form?
2. Why freeze the first 3 BART decoder layers? What is the impact of different freezing strategies or fine-tuning more/fewer layers?
3. Your power-law fits project substantial gains at 100k trials. How do you reconcile this with known long-term nonstationarities (electrode drift, scarring, etc.) that would violate the stationary-distribution assumption?
4. In the NHS module, how sensitive are results to the FiLM modulation strength?
5. Could you quantify attention alignment, for example, by correlating peak cross-attention with speech envelope events?
6. Given that inference is much faster than two-stage systems, have you considered beam search or other techniques to generate multiple hypotheses for rescoring? This might close the WER gap in Table1.

---

> ### Author Response · Authors · 2025-11-28
>
> We thank the reviewer for the careful reading and detailed suggestions.
>
> 1. Scaling laws and extrapolations (Weakness 1 and 2, Q1, Q3).
> We agree that the original scaling analysis was under-specified and that stationarity is a strong assumption for intracortical recordings. In the revised manuscript we have:
> 1) Re-ran the scaling experiments with one model variant (seq2seq + MFCC + BART) but 5 seeds per dataset fraction, and now visualise bootstrap 95% confidence intervals for both PER and WER in Fig. 2 (left),
> 2) Fitted the power law on the multi-seed mean curve and report bootstrap CIs over the fitted parameters and extrapolated curves in Fig. 2 (right) and Appendix D, making the quantitative uncertainty explicit,
> 3) For sensitivity to the functional form, we tested including vs. excluding the smallest 5% fraction (pre-asymptotic regime) - this choice shifts the constants slightly but preserves the key monotonic, approximately log–log linear trend between supplied data and performance (Table 6 in Appendix. D).
>
> Finally, we fully agree that the numerical projections at 20/100k trials should not be taken as literal performance guarantees; instead, we see them as optimistic lower-bound error estimates under stable recording conditions and discuss violations of stationarity due to drift, encapsulation, and electrode degradation (revised Section 3.1). Our intended takeaway is that the model exhibits clean scaling behavior and that there is substantial headroom if larger, reasonably stable intracortical corpora becomes available.
>
> 2. Attention analysis and interpretability (Weakness 2, Q5).
>
> We agree that relying on a single “representative” trial is not sufficient. In the revised paper we have:
> 1) Added in Section 3.4 a semi-quantitative measure of the pronounced temporal chunking pattern - each author was requested to visually inspect all the validation trails and assess whether the box structure is present or not. All authors individually confirmed existence of the encoder’s “temporal chunking” pattern in all 880 validation trials
> 2) Added two additional figures in the Appendix E showing attention maps across 20 randomly sampled validation trials and across four model variants and 15 trials, respectively. These panels illustrate that the “boxed” temporal chunking and reuse by both decoders appear broadly, regardless of the auxiliary objective, and are not idiosyncratic to a single example.
>
> While those additions don’t supply direct quantitative evidence for the observed patterns, they illustrate their prevalence. Lastly, when interpretation studies are considered, especially involving attention visualisation, they inherently rely on qualitative measures due to the nature of the evidence requiring complex, often non-automatable, visual interpretation.
>
> 3. Hyperparameter sensitivity (Weakness 4).
> During the architecture development it was found that the model dimensionality, num layers, dropout and learning rates were crucial for optimal performance. However, as both the hyperparameters and the model’s code were changing in parallel we are not able to provide numerical ablations from that period; in the interest of time we also didn’t perform them in this rebuttal period, choosing instead to focus on other requested experiments.
>
>
> 4. BART freezing strategy (Q2).
> We now discuss this in more detail in Appendix. C. We compared three setups for the BART decoder: (i) pretrained with first 3 layers frozen (our main setting), (ii) pretrained with all layers fine-tuned, and (iii) same architecture with random initialization. All three achieved very similar PER/WER, with the fully fine-tuned pretrained model giving slightly better WER and the randomly initialized decoder nearly matching it -  suggesting that in this data regime most useful linguistic structure can be learned from the supervised task, with pretraining adding only incremental gains. The partially frozen configuration was chosen as a balanced compromise between PER and WER.
>
> 5. NHS FiLM modulation strength (Q4).
> In the NHS module the final FiLM modulation strength is determined by the learnable “Gate and smooth mix” component, and not set by an external hyperparameter - therefore we are not able to test that ablation.
>
> 6. Multiple hypotheses and rescoring (Q6).
> We agree and thank the reviewer for this suggestion that is now implemented in the revised manuscript. The last paragraph in Section 2.2 1 and Appendix. B now describe a generation + rescoring pipeline where we: (i) generate candidate sentences using a mix of beam search and stochastic sampling, and (ii) rescore them using a linear combination of PER-based and phoneme-head scores plus an external LM score.
> This procedure reduces WER from 25.6% (greedy) to 19.4%, bringing our seq2seq approach to within 2 absolute WER points of the Willett WFST+LM system, while still maintaining substantially faster inference.

---

### Official Review · Reviewer_E9Q6 · 2025-11-07

**Soundness:** 2
**Presentation:** 2
**Contribution:** 1
**Rating:** 2
**Confidence:** 4

**Summary:**

The paper proposes a seq2seq Transformer that decodes intracortical signals directly into phonemes and words. It uses (i) a multitask setup with an auxiliary MFCC head, (ii) a frozen-part BART word decoder for language priors, and (iii) a day-specific “Neural Hammer & Scalpel (NHS)” transform (global affine + FiLM) to address across-day drift.

**Strengths:**

1) The paper tackles a relevant and challenging problem: the limited-data issue of human brain–computer interface datasets.
2) The paper includes clear, well-documented architecture and training details which helps reproducibility.

**Weaknesses:**

1) The novelty of the work is limited. The core contribution is a conventional seq2seq Transformer with (a) an auxiliary MFCC prediction head, (b) a partially frozen BART decoder, and (c) a day-wise affine+FiLM calibration (NHS). None of these ingredients are algorithmically new in ML; NHS is essentially per-day affine re-mix + FiLM gating. The work reads as careful engineering, not a conceptual advance.

2) All core results are reported on one intracortical participant from Willett et al. Moreover, the paper discards one of the implanted areas and uses only area 6v (128 channels), further narrowing scope. There is no cross-subject or cross-implant generalization. Therefore, conclusions about general utility are not supported.

3) The evaluation scope of the paper is narrow. It primarily contrasts variants of the authors’ own model and omits comparison with other baselines for intracortical or speech decoding  in Table 1 (e.g., transducer-based, CTC-Transformer, or hybrid pipelines). As a result, the contribution cannot be properly contextualized within the existing literature on brain signal-based speech decoding.

**Questions:**

Were the WFST + LM results recomputed under your preprocessing pipeline, or were they taken directly from Willett et al. (2023)?

---

> ### Author Response · Authors · 2025-11-28
>
> We thank the reviewer for the clear assessment and acknowledgement of tackling the important low-data regime challenge of human BCI.
>
> Weakness 1. We agree that the individual components of our architecture are not algorithmically new in general-purpose ML. Our claim is that their combination and systematic evaluation in the intracortical speech-decoding setting constitute a conceptual and empirical advance. A central goal of our work is to propose an alternative to the currently dominant way of handling the lack of temporal alignment between neural activity and speech labels. Framewise CTC is the prevailing solution in speech BCIs, but it comes with strong locality and conditional-independence assumptions that are inherently limiting. Our seq2seq approach removes these assumptions: the decoder conditions on the full neural context and on its own output history, and we show that this leads to substantially improved phoneme decoding on the utilised dataset, opening the door to further gains with more specialized Transformer variants.
>
> Beyond the core architecture, we show that multi-objective training improves data efficiency, directly addressing the low-data constraint characteristic of human BCI datasets. MFCC supervision improves PER despite severely degraded audio, indicating that weak acoustic cues act as a useful inductive bias for early neural representations. Adding a BART-based word head not only enables direct open-vocabulary decoding, but also nudges PER downward, demonstrating that word-level supervision positively impacts intermediate phoneme representations. Importantly, we demonstrate the efficacy of lightweight end-to-end neural-to-word decoding that contrasts with prevalent two-stage hybrid pipelines: our word decoder has ~100 million parameters (rather than LoRA-style adaptations of billion-parameter LMs from the end2end baseline), yet combined with generation+rescoring (added in the revised version, see Section 2.2 and Appendix B) it comes within 2 WER points of the Willett WFST+LM system, while offering substantially reduced inference time—crucial for prospective clinical deployment. Finally, our attention analyses provide an account of how a high-performing artificial system organizes and maps intracortical activity into speech units, offering a data-grounded hypothesis space for the brain’s representational strategies. While we appreciate that ICLR emphasizes algorithmic advances, the call for papers also explicitly lists important application areas and visualization/interpretation of learned representations as relevant themes; our work is intended to speak to these axes as well, fostering discussion about alternative modelling paths in the application-focused NeuroAI field.
>
> Weakness 2. We agree that the scope is limited to one participant and one cortical area, and we now highlight this more explicitly in the paper. Our goal is to provide a proof of concept that a seq2seq approach can challenge standard framewise CTC pipelines on a widely used public dataset, not to claim cross-subject generality. Extending the approach to multi-participant settings will require substantial architectural changes (e.g., subject adapters, time-relative calibration), which we view as important future work but could not execute within the rebuttal window. The restriction to area 6v follows both Willett et al. and our own preliminary experiments, where including area 44 did not improve performance while increasing compute; we now state this rationale clearly. To at least probe robustness within this constraint, we added held-out-day generalization experiments, which quantify how far the day-parametrized NHS transform can be extrapolated and where it fails (presented in Section 3.2)
>
> Weakness 3. For baselines, our choices were constrained by the lack of directly comparable methods that (i) report both phoneme- and word-level metrics and (ii) use compatible validation/test splits. This is particularly important because we study the interaction between acoustic, phoneme, and word tasks. We therefore focused on baselines that allow clean, controlled comparisons (original Willett RNN–CTC + LM, BGRU-Phone, and our RNN–CTC variants with a shared front-end and NHS). In the revised Results, we now explicitly discuss RNN-Transducer and Transformer–CTC work on this dataset, clarifying why their reported numbers are not directly comparable but still useful as architectural context.
>
> WFST + LM question. For both Willet and Feng baselines the PER and WER values in Table 1 are taken directly from respective publications.

---

### Meta-Review · Area_Chair_cWYE · 2025-12-29

**Summary:**

1. [E9Q6, Sx4q, 1HWu] The proposed architecture is a combination of known methods.
2. [E9Q6, Sx4q] Results are presented for only a single subject, so it is unclear how well the proposed model will generalize.
3. [E9Q6, Sx4q] The evaluation scope is limited, focused on ablations of the proposed model, some design choices ae not ablated, and more qualitative analysis would provide richer insights.
4. [1HWu] Model performance is not competitive with the baseline from Willett et al., 2023.
5. [cjht] The scaling analysis is based on single seeds and assumes that the neural data distribution is stationary, which is unlikely in practice.
6. [cjht] The analysis of attention patterns is limited and could be improved by making it more quantitative.
7. [cjht] The authors state that training stability and performance [are] highly sensitive to hyperparameters but do not illustrate the point with results.

**Reviewer Concerns:**

1. This concern is well addressed in the rebuttal. The authors agree that the components of the model are not new, but argue that their application to intracortical speech decoding constitutes a conceptual and empirical advance and that the use of multiple objectives is extremely important for BCI tasks where training data is necessarily very limited. I find the argument persuasive.
2. This concern is only partially addressed in the rebuttal and revision. The authors agree that the scope is limited and revised the paper to make this clearer. They also interpret the comment about cross-subject generalization to be a request for a model that can be trained on multiple participants, but I think the reviewers simply would have liked to see how well the model works on single-subject data from another subject, for example the dataset from Card et al., 2024. The closest the authors get to addressing this concern is their addition of experiments in the revision that perform held-out day ablations. This focus on only one subject is the most serious remaining deficiency of the paper.
3. This concern was addressed in the revision and rebuttal. The authors correctly point out that they are restricted to baselines where it is possible to report both phoneme error and word error rates, and they added discussion of other results, including explanations of why they are not directly comparable, to the revised paper. They explain that some design decisions were not ablated because they gave consistent phoneme error rate improvements in pilot experiments, and they added some more qualitative error analyses to the revision.
4. This concern is partially addressed in the rebuttal and revision. The authors agree that the Willett et al. model has better word error rate, but point out that their model has better phoneme error rate, can perform decoding faster, and added rescoring experiments that close the gap to the Willett et al. model somewhat.
5. This concern was addressed in the rebuttal and revision: the authors redid the scaling law experiments using a single model variant and five random seeds per dataset, and they updated the revision to clarify that the scaling law results represent a best-case scenario.
6. This concern was addressed in the rebuttal and revision: the authors manually verified that all 880 validation trials exhibited the blockwise attention pattern and they provided a random sampling of attention maps in an appendix.
7. This concern was not addressed in the revision and rebuttal, but I consider it to be a minor issue.

**Reviewer Scores:**

E9Q6 - I do not believe that this reviewer would have changed their score.
cjht - I do not believe that this reviewer would have changed their score.
Sx4q - I do not believe that this reviewer would have changed their score.
1HWu - I do not believe that this reviewer would have changed their score.

---

### Decision · Program_Chairs · 2026-01-26

Reject